# Influence of ESGC Indicators on Financial Performance of Listed Pharmaceutical Companies

**DOI:** 10.3390/ijerph18094556

**Published:** 2021-04-25

**Authors:** Alberto A. López-Toro, Eva María Sánchez-Teba, María Dolores Benítez-Márquez, Mercedes Rodríguez-Fernández

**Affiliations:** 1Department of Economics and Business Administration, Campus El Ejido, University of Málaga, 29071 Málaga, Spain; aalopez@uma.es (A.A.L.-T.); emsanchezteba@uma.es (E.M.S.-T.); 2Department of Applied Economics (Statistics and Econometrics), Campus El Ejido, University of Malaga, 29071 Málaga, Spain; bemarlo@uma.es

**Keywords:** ESGC indicators, controversies, financial performance, pharmaceutical industry, stakeholder theory, PLS-SEM

## Abstract

The pharmaceutical industry, concerned about the impact of its activity, has integrated responsible principles and practices with a view to improving its sustainable and financial performance. This study analyzes the relationship between environmental, social, governance, and controversy indicators and financial performance, measured through return on equity (ROA), return on assets (ROE), and Tobin’s Q, which are applied to the listed companies in the Nasdaq US Smart Pharmaceuticals Index. This index is composed of 30 international companies with a presence at the global level. All the data have been extracted from the Thomson Reuters database. The analysis was performed using structural equation modeling implemented with partial least squares. The results confirm the positive relationship between the construct composed of environmental, social, and governance (ESG) indicators and the aforementioned financial ratios. Additionally, a positive relationship of the controversy indicator with Tobin’s Q is supported. This suggests that the pharmaceutical multinationals focus their investments in sustainability on ESG and pay attention to controversies to boost the visibility of the company and thus increase its value. These conclusions confirm that investing in ESG is a profitable strategy. It is also relevant for managers as it increases the profits and the market value of multinational pharmaceutical companies.

## 1. Introduction

Corporate governance, the environment, and society indirectly affect the performance and activity of companies [1]. It is evident that the importance of this issue has increased in recent years at the international level, as shown by the number of both institutional and business initiatives that focus on promoting it and giving it visibility (the World Business Council for Sustainable Development, ISO26000, Global Compact, Global Reporting Initiative, among others). Furthermore, the interrelationship between CSR and sustainable development has more recently been highlighted in the 2030 agenda and the alignment of companies with the Sustainable Development Goals [2]. In this sense, CSR is the contribution of companies to sustainable development [3], making social and corporate benefits compatible [4] and maximizing stakeholder satisfaction [5]. To this end, CSR voluntarily incorporates social, environmental, and good governance concerns that are part of the company’s strategy into all its operations [3]. However, enterprises need their contributions to sustainable development to be visible and valued on the stock exchange in order to improve their competitiveness and financial performance.

Given the impact of pharmaceuticals on sustainable development, it is vitally important that the industry integrates sustainable green principles and practices sustainability at the management level [6]. Pharmaceutical companies incorporate sustainability concerns into their strategies and mission statements in different ways [7]. However, despite the increasing importance given to CSR by multinational pharmaceutical companies, there is no clear and standardized definition of CSR in global health that can serve as a benchmark for the reformulation of their CSR strategies [8]. Moreover, the pressure on pharmaceutical firms to respond to the challenges posed by CSR is increasing [9,10]. However, the resources that pharma allocates to research and development hinder the development of CSR strategies [11]. In this line, a study by Demir and Min [12] on a selection of leading pharmaceutical multinationals in the sector finds that sustainability reports show very mature and standardized areas, such as environment and work relations, etc. Conversely, other fields that correspond to particularly sensitive issues, such as rights of citizenship or supply networks, are not mature. In this sense, pharmaceutical companies should consider what their value proposition to stakeholders is [7]. They determine the best CSR strategies to add value and satisfy the needs of stakeholders. On the one hand, this fact satisfies the different opinions that consumers, employees, shareholders, health professionals, scientists, patient associations, the media, regulators, and non-governmental organizations (NGOs) have of the pharmaceutical industry. Society is suspicious of the excessive price of pharmaceutical products, irresponsible animal testing, and the lack of access to pharmaceutical products, especially in developing countries. On the other hand, reduced profit margins due to the pressure posed by the demands for a decreased patent duration and safer products result in the pharmaceutical industry concentrating its activity in fewer large companies [7].

Pharmaceuticals are essential for human, animal, and plant health, but they have a significant negative effect on the environment, as found in recent studies [13,14]. Pharmaceutical production is associated with a high consumption of raw materials, solvents and energy, producing a high volume of waste, and, in many cases, heavily polluting, and thus affecting health through water pollution [15]. Furthermore, once consumed and excreted, pharmaceutical products can enter the environment and be found in groundwater and soil, with harmful effects on aquatic organisms [16]. That is why the pharmaceutical industry has increased its concern for the environment and, consequently, has adopted sustainable practices [17].

In this sense, Milanesi et al. [18] point out that the pharmaceutical industry is especially interested in the “green” supply function management (production, materials, human resources, logistics, etc.). Furthermore, pharmaceutical enterprises’ management of waste, their participation in the health system in an economical and sustainable manner, and the analysis of developed and emerging markets are little-explored research lines, according to these authors [18].

Since the 1960s, many studies have sought to establish a positive relationship between the environmental, social, and governance (ESG) criteria and the financial performance of the companies [19]. Few studies focus on showing this relationship through joint ESG and financial performance indicators of companies [19,20,21,22,23,24,25]. Only one jointly incorporates the so-called “ESG controversies” indicator (hereinafter, we refer to this as “controversies” for easy reading) [26], and no study focuses on the pharmaceutical industry sector. The reason why we add the controversies to our analysis is due to their impact on business performance. Public information about sustainable actions, appearing in rankings of controversial companies or news in the financial press have a significant influence on companies’ financial performance.

### 1.1. Stakeholder Theory

A bibliometric study by Rodríguez-Fernández et al. [27] concluded that the most commonly used theories for the analysis of the relationships between CSR, CG, and FP are, (from most to least commonly used), stakeholder theory, followed by agency theory, and institutional theory. This is the reason why we have focused this study on the first of these theories.

Based on the stakeholder theory, the competencies of the components of the boards of directors include both balancing the interests, of the company and those of the interest groups as well as obtaining the maximum level of profitability for the shareholder. Their correct decisions will increase the levels of performance and, consequently, the establishment of an adequate social responsibility policy. The theory of stakeholders and the consequent interest of companies in improving well-being explains CSR’s management [28]. Hence, this theory points out the need for companies to actively participate in CSR practices, providing benefits to the society as a whole. Among the studies concerning stakeholder theory, we highlight those in [1,5,29,30,31,32,33,34].

The common denominator in the above contributions is the holistic nature of the theory and its applicability to the real enterprises. The authors include various interest groups, such as members of the boards of directors, employees, clients, or users of the services provided, suppliers, and all those external agents who affect or are affected by a company’s decisions [27].

According to Brammer and Millington [28], a company’s financial performance is influenced by the satisfaction of its stakeholders. Donaldson and Preston [5] indicate that satisfying the needs of stakeholders should result in better economic performance measures. Stakeholder theory holds that it is in the best interests of companies to operate in accordance with the needs of all stakeholders. In this sense, ESG indicators measure a company’s concern for the needs of its stakeholders and their satisfaction. ESG pillars meet the information needs that stakeholders have regarding environmental, social, and governance initiatives, as well as controversies.

In health and technology, many authors rely on stakeholder theory as a framework on which to base their analyses [35,36,37,38,39,40,41], giving the theory validity not only in terms of its formal content, but also at an empirical level. Our study is based on stakeholder theory, as it takes into account the participation and interests of stakeholders through the ESG indicators.

The search for the relationship between the ESG criteria and corporate financial performance dates back to the early 1970s [21]. Thus, Carroll and Shabana [42] refer to the financial performance as a reason for why companies to apply CSR strategies and policies. Papers in this area report mostly positive relationships [19,20,21,22,23,24,25,26,27,43,44,45,46]. The study of Margolis et al. [47] reports, in general terms, a weak positive effect between social and financial performance, finding that the association between charitable organizations and environmental performance is stronger. Other studies, such as that in [48], show clear empirical evidence of a positive correlation between companies’ social and financial performance, indicating that “Good ethics is good business”. Even so, Orlitzky et al. [49] consider that the results of this research should be treated with caution and propose a more rigorous alternative methodology.

Studies such as that of Wood [50] point to the need to focus research on stakeholders and society. More specifically, Wagner and Schaltegger [51] find that the social and governance indicators have a positive influence on financial performance for European industrial listed companies, while the environmental performance of the enterprises did not show a significant effect. Porter et al. [52] show that improving the company’s environmental performance can lead to better economic or financial results. In turn, Ambec and Lanoie [53] study the pathways that can lead to increased or decreased profitability ratios due to environmental practices.

Beyond pure economic benefit [54] and compliance with legislation, companies should become involved in environmental actions as an essential part of their strategy. As a consequence of the latter, they obtain long-term economic benefits [55], derived from the achievement of competitive advantages gained [52,56]. This improvement caused by the adoption of environmental initiatives [57] can also reduce resources for investment that would decrease their competitiveness [58], thus taxing short-term economic benefits [59]. Financial performance is diminished due to the diversion of resources to environmental investments [60,61,62]. In contrast with this idea, Dalal and Thaker [63] studied Indian companies included on the NSE 100 ESG Index database for the period of 2015–2017, showing that a good corporate ESG performance improves financial results. Paolone et al. [64] consider market share to be a function of the ESG pillars score, and they use the ESG score as a proxy of the fundamental strategy of corporate sustainability.

### 1.2. Hypotheses

The literature on whether CSR activities create value for a company’s shareholders, or if, on the contrary, they focus too much on stakeholders, and reduce the value, is divided [65,66,67,68]. The complexity of the mechanism through which CSR affects a company’s value and proposes an indirect link is stated by Servaes and Tamayo [69] based on Barnett’s idea [70]. The latter authors concluded that companies with a high level of consumer awareness could increase their value by increasing CSR, if this was accompanied by a good reputation. For Paolone et al. [64], ESG metrics are conducive to a high marketing performance, highlighting the high impact on the governance pillar as an essential determinant for ensuring a higher level of market performance. In parallel, while CSR activities have a negligible or negative impact on company value, the costs sometimes exceed the benefits due to a low advertising intensity [69].

To support the previous arguments, Aouadi and Marsat [71] observe a significantly direct negative effect of the controversy indicator on ROA performance, whereas Tobin’s Q performance is not directly and significantly affected.

Other authors, such as Churet and Eccles [72], found report a strong relationship between integrated reporting and the quality of the management of ESG quality of management in certain sectors, notably healthcare. Similarly, Sharabati’s work [73], another study of CEO’s perceptions of pharmaceutical companies in Jordan, shows the existing correlation between the CSR pillars and business performance. It adds that CSR’s effectiveness is justified by investing in the management of stakeholders’ management, such as customers, corporate governance, investors, and activists, creating positive relationships that enhance a company’s reputation and profitability. Ferrero-Ferrero et al. [21] emphasize that the companies with the greatest interdimensional consistency of the ESG have greater financial success.

For this reason, and given the influence that the industrial sector can have, the objective of this work is to show the relationship that exists, on the one hand, between the ESG construct and the financial performance indicators in the pharmaceutical industry and, on the other hand, between the controversies and the aforementioned financial indicators. ESG indicators are used to measure and understand CSR activities in the company [64]. Additionally, the valuation of ESG pillars jointly seems to be employed in most of the literature, as investors value them differently [74]. Still, there are doubts about the reliability of information provided by multinational companies through sustainability reports and ESG indicators [12]. For this purpose, a construct is proposed that includes the ESG indicators, and the controversy indicator is taken as a control variable. All of the indicators are regressors in the proposed model, and the indicators of financial performance in the pharmaceutical industry sector (ROA, ROE, and Tobin’s Q), are considered dependent variables.

This general objective is specified in a confirmatory–exploratory analysis of the following six hypotheses:

**Hypotheses** **1** **(H1).**
*The indicator controversies (CONT) influences ROA.*


**Hypotheses** **2** **(H2).**
*The indicator controversies (CONT) affects ROE.*


**Hypotheses** **3** **(H3).**
*The indicator controversies (CONT) positively influences Tobin’s Q.*


**Hypotheses** **4** **(H4).**
*The construct ESG positively influences ROA.*


**Hypotheses** **5** **(H5).**
*The construct ESG positively affects ROE.*


**Hypotheses** **6** **(H6).**
*The construct ESG positively influences Tobin’s Q.*


Hypotheses H1 and H2 are exploratory, so they have not been assigned any sign (positive or negative).

## 2. Materials and Methods

### 2.1. Data

The data are secondary and belong to the year 2018 database, which is restricted to the pharmaceutical sector and concerns pillar scores: governance (P1_GOV), environment (P2_ENV), social (P3_SOC), and controversy (P4_CONT) from the Thomson Reuters database. The pillar scale was previously established [75]. Additionally, referring to year 2019, three ratios of the aforementioned selected pharmaceutical companies are considered in the model: the Tobin’s Q for year 2019 (TOBIN19), return on equity for year 2019 (ROE19), and return on assets for year 2019 (ROA19). A total of seven indicators (four pillars and three accounting ratios) were considered for 25 companies.

The data for the ESG indicators and controversies are from the year 2018, and the financial performance indicators are from the year 2019, with a time-lapse of one year to note their influence [20,24]. The ESG indicators and ESG controversies are key aspects of the strategic direction of any company, and in the cases that concerns us are companies that belongs to the New York Stock Exchange, specifically those in the Nasdaq US Smart Pharmaceuticals Index (NQSSPH), which is composed of 30 international companies with a presence at the global level (see Table A1 of Appendix A). The conversion of the text data (with letters representing ordinal scales) into numerical values is conducted using by the mean of the intervals proposed by Thomson Reuters (see [75]). The aspects that we include in the environmental indicator include resources, emissions, and innovation. The social indicator includes the responsibility for production, the community, personnel, and human rights. Management, stockholders, and CSR strategy are included in the governance indicators. Controversies indicator compiles a list of 23 measures, including personnel, responsibility for production, stockholders, human rights, the community, management, and the use of resources.

As for the financial measures, the ROE variable is the outcome of dividing operating profit before depreciation and provisions by equity. Meanwhile, the value obtained by the division of the operating profit before discounting depreciation and provisions is the ROA variable. Tobin’s Q is the ratio resulting from dividing a company’s market value by the assets’ replacement cost.

Some indicators show outliers derived from the box-plot analysis using SPSS v25, but due to the size of the companies’ population, they have been included in the study. Additionally, enterprises with more than 10% of missing items were eliminated from the analysis, with three in total. One missing value was replaced by the corresponding mean. From a total of 30 companies, 25 were included after removing the outliers. A PLS-SEM model is proposed to study the effects of the pillars (p1, p2, p3, and p4) on the financial indicators (ROA19, ROE19, and TOBIN19) using the SmartPLS 3.3.3 software [76].

### 2.2. Confirmatory Model

Based on this study’s purpose, the initial model proposed is illustrated in Figure 1 and includes the ESG construct studied in the literature and the controversies indicator (CONT) as a control variable (a single-item construct).

### 2.3. PLS-SEM

SEM is a statistical method to assess a model, which simultaneously includes a latent variable and indicators for all the relevant relationships. Some of the reasons for using PLS-SEM, as pointed out by Cepeda et al. [77], among other expert scholars, include the complexity of certain other models, their use of formative constructs when the constructs are considered to be composites, and the small population size of their samples, which is due to their deletion of outliers and missing outliers (and they cannot fulfil the minimum required sample size). The sample size of 25 enterprises does not meet the minimum threshold of more than 100 recommended by Reinartz [78]. However, it does meet the minimum proposed by other earlier authors [79,80], who state that a minimum sample equals to the result of ten times multiplied by the largest number of arrows receiving a dependent latent variable in the nomogram of a structural model; in this case, 2 paths pointed to each ratio (Figure 1).

Two steps must be followed in the PLS-SEM assessment: on the one hand, the measurement model; and, on the other hand, the structural model. In the first step, the measurement model is estimated as the compounds in mode A. The rest were single-construct. The item reliability is first checked to guarantee that the items associated with a construct require values higher than 0.7. Additionally, the internal consistency reliability is evaluated through the composite reliability (CR) [81], Cronbach’s alpha [82], and Dijkstra–Henseler rho (ρA), which should be between 0.7 and 0.95. Some authors claim that values of reliability of 0.7 may be sufficient in research [83]. The convergent validity is measured by the average variance extracted (AVE), which should be equal to or over 0.50, meaning that the construct explains a percentage equal to or over 50% of the indicators’ variance.

In the second step, the significance and the magnitude of the standardized beta-path coefficients, serious problems of collinearity detected by variance inflation factor (VIF), the determination coefficient, R^2^ and the adjusted R^2^, the Stone-Geiser Q^2^ [84,85], and the effect size (f^2^) are analyzed in order assess to obtain the inner model’s assessment. According to Chin [86], a substantial, moderate, and weak predictive power corresponding to R^2^ values of 0.67, 0.33, and 0.19 should be considered. The Q^2^ value should be over the threshold 0.5 for the model to be considered a predictive model [86].

Moreover, the higher the value of the beta coefficient is, the higher the path’s relevance or the stronger is the impact. The beta-paths’ significance was assessed using the bootstrapping procedure of 5000 subsamples of a size of 25 via *p*‑value or t-statistics [87]. The evaluation of the change produced in the value of R^2^ by eliminating an exogenous latent variable from the PLS-SEM model can be used to point out whether the deleted latent variable represents a substantial effect on the endogenous construct. This evaluation of R^2^ change determines the effect size (f^2^). The effect size assesses the exogenous variable’s influence on the structural model’s predictive power. Several authors [77,88,89] recommended a systematic way to examine the PLS-SEM, and this process is followed in this case.

## 3. Results

### 3.1. Descriptive Analysis

The descriptive measures and the model are displayed in Table 1. The excess of kurtosis and skewness indicate the non-normality of the items.

### 3.2. Measurement Model

Table 2 displays the individual reliability of the items, the reliability of the latent variable, and the convergent validity. Following the values of these criteria concerning the quality of the measurement model, the quality of the ESG is supported. The rest are simple-item constructs.

The Fornell–Larcker approach [90] is used to assess the discriminant validity, which is satisfactory for all latent variables and meanings that construct measures different concepts. Additionally, another applied criterion is the Heterotrait–Monotrait (HTMT) ratio, and the results showed values below 0.85, which is below the recommended threshold (0.85 or 0.90) [91]. The outcomes for both criteria are shown in Table 3.

### 3.3. Structural Model

The coefficient of determination, R^2^, the adjusted R^2^, and the Stone‑Geisser’s Q^2^ are measures of the predictive power. The R^2^ values of ROA19, ROE19, and TOBIN19 were moderate (0.219, 0.193, and 0.191, respectively). In this case, the Q^2^ criterion of Stone-Geisser is lower than that of the established threshold Q^2^, which is >0.5, so the models were not considered to be predictive models (Table 4).

All of the f^2^ values fluctuate between 0.004 and 0.231. In this case, the effect size of ESG when explaining ROA19 and TOBIN19 are weak (0.140 and 0.149, respectively), and it is moderate when explaining ROE19 (0.155), following Cohen [92]. However, the f^2^ of CONT when explaining ROA19 (0.013), and ROE19 (0.004) is weak, and it is moderate when explaining TOBIN19 (0.231).

According to Hair et al. [88], all the VIF values are below the cutoff of 3.3, resulting in no problematic collinearity (Table 4). Figure 2 shows the estimated path model.

In Table 5 the estimated beta coefficients (in this case, direct effects) and the results of the tests of the proposed hypotheses are displayed.

Our results show positive relationships between ESG and ROA19, indicating that ESG positively influences ROE19 and affects Tobin’s Q. Meanwhile, the controversies are only supported in the relationship with Tobin’s Q. All these beta coefficients are very similar. Moreover, it is striking that ESG controversies only positively affect Tobin’s Q, and the effect is more significant than that of ESG and the f^2^.

## 4. Discussion

The correct methodology to employ in this context may be the subject of some disputation. The PLS-SEM was adopted based on the idea of the ESG construct as a theoretical concept. We considered the ESG construct as a “design construct” in line with Henseler [93,94], who pointed out that a better modelling can be achieved through composites, and PLS-SEM is therefore recommended. Another consideration that can be misunderstood is the size sample. Specifically, in studies conducted by experts in PLS-SEM, it is indicated that in cases of a population with a small size, it is reasonable to collect a small sample (fulfilling the minimum required sample size, as mentioned above in the method section) according to Cepeda [77]. The population of the mentioned Pharma companies belonging to the stock exchange is composed of 30 companies. However, the valid size sample is composed of 25 companies, after deleting companies with more than 10% of indicators with missing data, which are not considered in the study, and after removing extreme outliers. Finally, because ESG is composed of three indicators, the construct is reflectively related to the pillar indicators.

The ESG pillars of the pharmaceutical industry show a medium score, indicating that concern for social, environmental, and governance should be higher, and that the commitment to sustainability could be increased. This concern makes sense, since a failure to manage the ESG pillars generates stakeholder dissatisfaction with a company’s consequent damage, thus affecting its financial performance.

In general terms, there is a lack of consensus and a disparity in the literature reviewed. The results obtained in our study for multinationals in the pharmaceutical industry have been confirmed by other researchers in various sectors [19,20,21,22,23,24,25,26,27,43,44,45,46]. Furthermore, El Ghoul et al. [25] found a significant positive influence on financial performance. A review on the relationship between ESG development and financial performance shows that almost 90% of the studies found a non-negative relationship [19]. The study conducted by Chelawat and Trivedi [46] reports a good performance in terms of the environmental, social and corporate governance pillars measured through ESG indicators, with a positive impact on financial performance, as evaluated through ROA, ROE, and Tobin’s Q. In addition, Rodríguez-Fernández et al. [26] report a positive association between ESG pillars and financial performance in listed travel and leisure companies.

Likewise, the literature review confirms that this financial performance is positive and stable over time [19,43,44]. In addition, companies with a stronger relationship adopt a long-term value-oriented approach [51]. While our study shows a positive result, albeit for only one year, we found that multinational companies in the pharmaceutical sector adopted a long-term value-oriented approach. Considering ESG activities as an investment, a positive and stable relationship is expected.

Focusing on the value of companies measured by Tobin’s Q, which is related to the ESG construct, there are different findings in the literature. Thus, Hemple [14] shows, in companies belonging to the S&P index, that the performance of the ESG construct has a positive impact on the ROA, but it does not impact Tobin´s Q. Similarly, in the pharmaceutical industry in Jordan, Omar and Zallom [95] analyzed the possible effects on CSR and the relationship with the company’s market value, through Tobin´s Q by using multiple regression, and they concluded that environmental, community, and products activities did not affect the pharmaceutical industry. However, Mahapatra et al. [96] found that financial performance relates positively to ESG success in Indian companies. This highlights the possible differences between a company’s accounting valuations (Tobin’s Q) and its financial results (ROA and ROE). These discrepancies could be justified by the size of the company, the visibility to which it is exposed, and its region or country of origin [71]. Mahapatra et al. [96] also point out that a higher company value could be attributed to a higher score for companies that experience controversies, are large (multinationals), and are located in countries that enjoy a greater press freedom. According to these authors, this would show that the relationship between controversies and company value, according to Tobin´s Q, depends largely on investor perceptions about a company’s performance. Additionally, greater visibility affects the link between social development and enterprise value [71].

As mentioned above, the type of industry and size of companies can influence this relationship. Focusing on the pharmaceutical industry, a case study of a multinational corporation has been published that corroborates the contribution of CSR and sustainability to the improvement of companies’ financial performance [97]. This work suggests that profitability is sustained over the long term if financial performance is integrated with social and environmental objectives in the company’s strategic direction to benefit shareholders, consumers, society, and the community.

For their part, Min et al. [7] conclude, in a study of the perception of chief executive officers (CEO) in the pharmaceutical industry, that CSR adds value to a company’s financial performance, regardless of its size, and should be considered as a long-term investment.

Therefore, the different contexts (regions, portfolio and nonportfolio studies, investment in ESG, emerging markets, corporate bonds, and ecological real estate) could justify the academics’ different results. These factors may determine the existence, the sign, and the magnitude of the relationship. In this sense, the study of Cek and Eyupoglu [20] supports that for companies in different regions, the results can vary according to the objectives pursued.

A study of 365 listed companies from emerging countries analyzed the relationship between ESG pillars and financial performance in sensitive industries [98]. While the results reject the ESG relationship with financial performance, they show a positive effect between environmental performance and financial development. This indicates that while companies in sensitive industries are at increased risk [99], they are also subject to stronger control and often achieve a higher financial performance [100]. That is, sensitive and risky industries carry out increased ESG practices to legitimize their operations and increase their visibility [101,102,103].

Additionally, Chelawat and Trivedi [46] examined the effects of Indian companies’ ESG performance on financial results, indicating that a good ESG performance improves financial results. With the same approach, the work of Andersen and Dejoy [104], through factorial analysis of variance, indicates that the size of the variables, industry, risk, and research and development expenses must be controlled in order to adequately specify the model. More specifically, in Baron [105], it was found that investment in R&D is an important determinant of companies’ financial performance, and that, if we take this into account, CSR (parallel to ESG construct) has a neutral impact on financial performance. Finally, according to the authors, in the pharmaceutical industry, investment in R&D (such as vaccines) should be considered a key issue for this sector, as it is decisive for determining financial performance and productivity. Following Baron [105] and Hillman and Keim [106], the ESG indicators do not consider the type of CSR that the company is developing (altruistic or strategic), which can have a very different effect on profitability. This highlights the complexity of the relationship between CSR and financial performance and the need to analyze the mechanisms of that relationship.

Another set of results refers to the positive relationship between the controversies and market value. The incorporation of the controversy pillar as a control variable is an important novelty in this study. This relationship confirms the findings of Aouadi and Marsat [71], who state that this surprising relationship could be attributed to high-performance companies (with high ROA), larger companies (multinationals), and those located in countries that enjoy a greater press freedom, which is the case for the multinational pharmaceutical companies analyzed in our study. On the contrary, Siew et al. [107] note that most listed construction companies exhibit, on the one hand, low levels of information, and on the other hand, a weak correlation between financial performance and ESG scores. These studies align with those that point out that CSR activities have a negligible or negative impact on company value for those companies with low advertising spending [69].

The reviewed studies support the view that controversies increase a company’s visibility, which would positively affect the company’s social development score and its market value, in the case of large companies with high returns [71], which aligns with the findings of our study. Our results suggest that investments in sustainability should focus on the three ESG pillars together, with controversies acting as a control variable for pharmaceutical multinationals. Additionally, controversies enhance the positive effect on financial performance (measured by ROA and ROE) and a company’s value (measured by Tobin’s Q).

## 5. Conclusions

This analysis responds to the lack of studies on how ESG indicators relate to financial performance, in the case of the multinationals of the pharmaceutical industry. The main contribution of the work is that it supports the influence of the ESG pillars on the financial performance in the pharmaceutical sector. In other words, the aim of the study is to demonstrate that pharmaceutical companies can contribute to sustainable development and to their financial performance. This will help pharmaceutical firms to question their value creation approach to stakeholders and thus contribute to the formulation of their CSR strategies. Stakeholder theory supports our theoretical framework by taking into account all actors involved in the functioning of a company and in the development of sustainable strategies.

This research concludes that the sustainability, as measured through ESG indicators, is positively related to companies’ financial performance (ROA, ROE, and Tobin’s Q). The study focuses on multinationals in the pharmaceutical industry.

This paper also exhibits a positive relationship of the controversies pillar with the market value (Tobin’s Q), not with the financial performance (ROA and ROE). These conclusions confirm that investing in ESG is a profitable strategy, which is relevant for managers as it increases the profits and the market value of multinational pharmaceutical companies. Likewise, policymakers and international regulations should pay more attention to non-financial performance, controlling the quantity and quality of information on ESG pillars in companies and thus ensuring transparency and reliability in their measurements.

One of the main implications for the management of pharmaceutical companies is the orientation of strategic decisions concerning CSR and sustainable development. By taking ESG measurements into account, companies can transparently meet the needs of stakeholders, while contributing to sustainable business development. Likewise, policymakers and international regulations should pay more attention to non-financial performance, controlling the quantity and quality of information on ESG pillars in companies and thus ensuring transparency and reliability in their measurements.

There are some limitations of this study, which indicate the need for the further development of the relationship between ESG indicators and financial performance. Firstly, it focuses on data for the period between 2018 and 2019, and a longer time frame is therefore required to verify that the results are being met. Secondly, it is a sectoral study and focuses on multinational companies in the pharmaceutical field, so it cannot be extended to all sectors or the entire pharmaceutical industry. Finally, it can be expected that the results obtained are influenced by other factors, such as the country or region, and multinationals may have different strategic objectives in terms of sustainability or investment in R&D.

This issue should be investigated in greater depth, posing new questions concerning, for example, the effect of specific categories of controversies on financial performance or the impact of CSR (altruistic or strategic) on the score for ESG indicators. Future research will include an analysis of productivity based on the results for drug patents, drug approvals, and drug innovation of these 30 companies, as we are convinced that productivity is related to ESG indicators.

## Figures and Tables

**Figure 1 ijerph-18-04556-f001:**
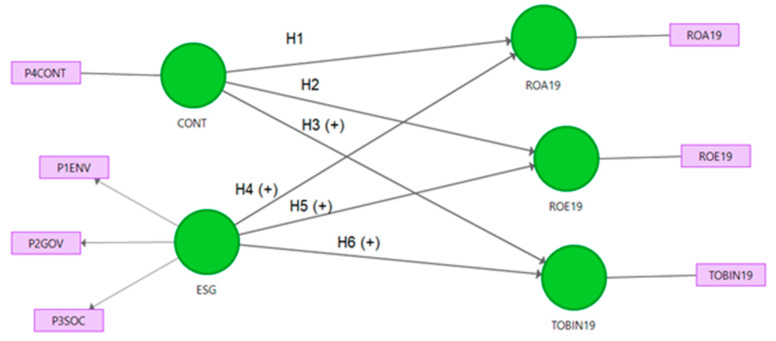
The theoretical research model.

**Figure 2 ijerph-18-04556-f002:**
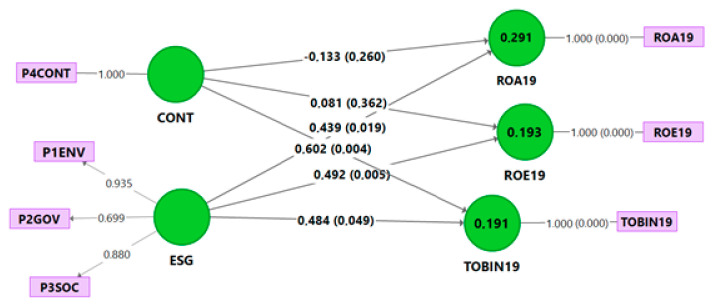
Estimated model (direct effect). Source: [76]. Note: Values in brackets refer to *p*-values.

**Table 1 ijerph-18-04556-t001:** Descriptive measures of the indicators.

Indicators	No.	Missing	Mean	Median	Min.	Max.	Standard Deviation	Excess Kurtosis	Skewness
P1ENV	1	0	0.343	0.29	0.04	0.96	0.317	−1.322	0.499
P2GOV	2	0	0.54	0.54	0.04	0.96	0.276	−1.283	0.024
P3SOC	3	0	0.481	0.46	0.21	0.87	0.208	−1.099	0.26
P4CONT	4	0	0.344	0.21	0.04	0.96	0.333	−1.332	0.526
ROA19	5	0	−7.754	1.31	−58.36	13.68	23.095	0.182	−1.254
ROE19	6	0	−23.043	2.55	−332.94	74.59	84.323	6.693	−2.359
TOBIN19	7	0	5.223	3.94	0.46	18.77	4.294	3.782	1.934

Note: Sample size is 25 companies. No., Number of items. Excess kurtosis from value 1. P1ENV = Pillar1 environment, P2GOV = Pillar 2 governance, P3SOC = Pillar 3 social, P4CONT = Pillar 4 controversies. Source: SmartPLS [76].

**Table 2 ijerph-18-04556-t002:** Reliability and validity assessment.

Indicators/	Loadings > 0.7			
Construct	ESG	CA > 0.7	CR > 0.7	AVE > 0.5
ESG		0.818	0.911	0.880
P1ENV	0.935			
P2GOV	0.699			
P3SOC	0.880			
P4CONT		1	1	1
ROA19		1	1	1
ROE19		1	1	1
TOBIN19		1	1	1

Note: CA, Cronbach’s alpha; CR, composite reliability; AVE, average variance extracted. Source: SmartPLS [76].

**Table 3 ijerph-18-04556-t003:** Fornell and Larcker, and Heterotrait–Monotrait criteria to assess discriminant validity.

Indicator/Construct	1. CONT	2. ESG	3. ROA19	4. ROE19	5. TOBIN19
1. CONT	1	0.710	0.438	0.262	0.265
2. ESG	−0.696	**0.844**	0.483	0.410	0.129
3. ROA19	−0.438	0.531	**1**	0.851	0.094
4. ROE19	−0.262	0.436	0.851	**1**	0.006
5. TOBIN19	0.265	0.065	−0.094	0.006	**1**

Note: The square of the average variance extracted (AVE) in bold in the diagonal. The latent variable’s inter-correlations are located under the diagonal, and the Heterotrait–Monotrait (HTMT) values are positioned over the diagonal.

**Table 4 ijerph-18-04556-t004:** Assessment of the quality of the inner model.

Indicator or Construct	R^2^	R^2^ Adjusted	Q^2^	f^2^		VIF(<3.3)		
				CONT	ESG	3	4	5
1. ESG						1.942	1.942	1.942
2. CONT						1.942	1.942	1.942
3. ROA19	0.291	0.227	0.238	0.013	0.140			
4. ROE19	0.193	0.120	0.117	0.004	0.155			
5. TOBIN19	0.191	0.117	0.112	0.231	0.149			

Note: SmartPLS [76].

**Table 5 ijerph-18-04556-t005:** Path coefficients, significance and hypotheses outcomes.

Hyp	Paths	Beta Path β	f^2^	Signif	*t*-Statistics	*p*‑Value	Supported/Rejected
**H1**	CONT -> ROA19	−0.133	0.013	NS	0.644	0.260	H1 rejected
**H2**	CONT -> ROE19	0.081	0.004	NS	0.353	0.362	H2 rejected
**H3**	CONT -> TOBIN19	0.602	0.231	**	2.634	0.004	H3 supported
**H4**	ESG -> ROA19	0.439	0.140	**	2.070	0.019	H4 supported
**H5**	ESG -> ROE19	0.492	0.155	**	2.582	0.005	H5 supported
**H6**	ESG -> TOBIN19	0.484	0.149	**	1.651	0.049	H6 supported

Note. Hyp, Hypothesis; Signif, statistically significant; NS, not significant; ** level of significance of 0.05.

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
