# Peer review of "Influence of ESGC Indicators on Financial Performance of Listed Pharmaceutical Companies"

_ijerph, 2021, doi:10.3390/ijerph18094556_

Round 1
Reviewer 1 Report
This manuscript deals with an important topic of the relationship between the financial performances of 30 international companies of the pharmaceutical sector and the Environmental, Social, Governance, and Controversies indicators. Despite the importance of the topic, but the manuscript has been written in a very confusing way. Additionally, several concerns to be addressed as follow:
1. A major concern of this study is the short time frame. To give a convincing and reliable conclusion of this study, it is highly recommended to extract the data for more than one year.
2. The introduction needs to be reorganized. For instance, the aim of the study has been repeated two times (lines 75-78 then lines 163-165).
3. In the material and methods section, several points should be transferred to the discussion section like the justification of the selection of SEM lines 22-273 should be transferred to the discussion section.
4. Results:
- Data in table 1 should be just mentioned in the text.
- The full term of all abbreviations in tables should be mentioned in the footnote.
5. The conclusion needs to be more concise. Also, lines 419-422 should be transferred to the discussion.
6. The way in writing the references in the text contains many errors. E.g. line 89: According to [28] should be Clarkson [28]. This error has been repeated many times throughout the manuscript especially lines 101-117.
7. Many references are outdated like 1, 48, 49,…etc. Please update.
8. There is a problem in using abbreviations throughout the manuscript. The full term must be introduced upon the first mentioning followed by its abbreviation in parentheses: From then on, the abbreviation must be used exclusively and throughout. E.g. in line 43: NGOs should be mentioned as non-governmental organizations (NGOs).
9. The language is highly recommended to be revised by a native English speaker.
Author Response
Attached I am sending the response to reviewers jointly. Yours sincerely, Mercedes Rodríguez-Fernández

Reviewer 2 Report
Paper has a really interesting point, however, it is missing a major contribution.
I like the topic and the execution of data. However, I have some concerns regarding the methodology .
1) Hypotheses are listed rather than discussed. What is the theoretical background of this study?
2) I am extremely confused about data and sample selection. Is this only for one year of data? Is there a time span? What is the variation in the sample?
3) There is no number of observations in table, hence, I don't understand the sample construction.
4) Please add stock performance as an alternative measure for shareholder wealth.
5) As a robustness test, please break down the components of ESG as E and S and G, and test them separately.
6) Results are listed, rather than discussed.
7) In Appendix A, some of those firms are not US firms. You should also create a binary variable if the company is located in the USA. Since USA requires more transparency, results could be different for US firms.
8) You can easily obtain drug patents/drug approvals/drug innovation data via online sources. Authors should also test the relationship between ESG and productivity. While ESGC may promote profitability and mitigate Tobin's Q issues, shareholder wealth (stock performance) and productivity (patents or approvals) are left out. Without those metrics, paper doesn't seem complete.
Author Response

(The authors gave the same response as above.)

Reviewer 3 Report
Major issues 1) Theoretical underpinning is weak as the paper heavily relies on one theory- stakeholder theory. There are some other relevant theories that must also be included in the literature review and discussion of analysis. For instance, see the following papers: Paolone, F., Cucari, N., Wu, J. and Tiscini, R. (2021), "How do ESG pillars impact firms’ marketing performance? A configurational analysis in the pharmaceutical sector", Journal of Business & Industrial Marketing, Vol. ahead-of-print No. ahead-of-print. https://doi.org/10.1108/JBIM-07-2020-0356 Nussbaum, A. (Sascha) K. (2009). Ethical Corporate Social Responsibility (CSR) and the Pharmaceutical Industry: A Happy Couple? Journal of Medical Marketing, 9(1), 67–76. https://doi.org/10.1057/jmm.2008.33 There are numerous definitions of the corporate social responsibility (CSR), none of which are cited by the authors. Instead, they have related CSR to sustainable development goals (which also needed a citation as these are United Nations promulgation). The following paper criticizes a common generic definition and seek a novel definition of CSR for global healthcare sector. The paper argues that CSR is not even understood in the same way across the pharmaceutical industry. Droppert, H., Bennett, S. Corporate social responsibility in global health: an exploratory study of multinational pharmaceutical firms. Global Health 11, 15 (2015). https://doi.org/10.1186/s12992-015-0100-5 The following paper suggest that the pharmaceutical companies still resort to selective disclosure techniques to highlight their achievements in areas where they feel more confident while leaving out others that can have potential negative consequences on the company. How reliable are then the ESG pillars used in this research? Demir, M. and Min, M. (2019), "Consistencies and discrepancies in corporate social responsibility reporting in the pharmaceutical industry", Sustainability Accounting, Management and Policy Journal, Vol. 10 No. 2, pp. 333-364. Structural Equation Model (SEM) is not suitable for a number of reasons including those mentioned and acknowledge by the authors. SEM approach lends itself for better use when the operationalization of a construct require use of latent, mediating and/or moderating variables. Use of survey data or mixed method approaches such as surveys and interviews are best suited for SEM. ESG data obtained from Refinitive Thomson Reuters Database is a text or character in nature, the authors did not explain in the method section, how these grades were converted into numeric values prior to use in SEM. I would recommend that the authors state specifically what are the major indicators of 30 international companies that shows their "global" presence. For instance, global sales volume, global assets, number of foreign subsidiaries, number of employees worldwide etc. I am afraid listing on the NYSE or NASDAQ and their tickers does not give away useful information about their global scale. I would recommend that the authors use a simple linear regression estimation approach and correlation (non-parametric such as Spearman) to test their hypotheses. Keep the financial performance variables ROA and Tobin's-Q ratio (t+1), include main variables, then lags (t-1) of the control variables total assets, leverage, growth, research and development expenses, capital expenditure, and lagged ROA to correct for serial correlation problems. Discussion section repeats some of the text from earlier sections, it would improve the flow of paper to limit such repetitions. The final section of the research papers usually have implications and recommendations, this paper has none of them.Author Response
Attached I am sending the response to reviewers jointly. Yours sincerely, Mercedes Rodríguez-Fernández

Round 2
Reviewer 1 Report
The authors adequately responded to all comments and performed the required modifications as directed.
Author Response
Thanks a lot for your help in the process of reviewing our work.
Best wishes,
The authors
Reviewer 2 Report
Thank you.
Author Response

(The authors gave the same response as above.)

Reviewer 3 Report
The authors have made changes to improve the quality of the paper. However, there is an error in section 2.1 Data; the three pillars are governance, environment and social. The authors used the word 'government'. Instead of using the word "financial development", it is better to use "financial performance, " commonly used in the literature.
Author Response
Dear Reviewer,
Thank you for your help in the whole process of reviewing our work. We have undertaken the minor changes indicated by you. Thus, we have changed the word “government” to “governance” and the word “development” to “performance”, as indicated by reviewer 3.
I have marked these changes in red in the final document.
If we need to make any other changes, please let us know.
Sincerely,
The authors